# Speed and Sustainability When Entering New Market

**Gwang Seok Kim *** and **Young Hoon Lee** 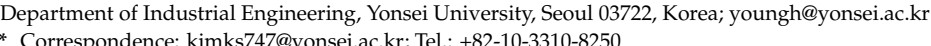

Department of Industrial Engineering, Yonsei University, Seoul 03722, Korea; youngh@yonsei.ac.kr
* Correspondence: kimks747@yonsei.ac.kr; Tel.: +82-10-3310-8250

**Abstract:** When constructing a factory to enter new markets, the optimal size to respond to demand is determined by the construction time. Hyundai Motor Company (Hyundai), on the other hand, standardizes the size of its factories to speed up the entry and response to demand. The Hyundai's entry mode, called SPEED, is modeled as a strategy. The strategy is evaluated of excellence with capacity expansion rules formalized, key parameters identified, and mathematical programming. The SPEED strategy is suited for market followers who want to enter a midscale or mature market in terms of business excellence and more sustainable throughout the factory's life cycle on the side of sustainability. Shorter construction times, as a result of the SPEED strategy, can help to prevent environmental damage while also standardization can increase job prospects for local workers.

**Keywords:** capacity expansion; standardization; speed; sustainability; Hyundai

## 1. Introduction

Global automakers, such as, Toyota, Volkswagen (VW), GM, Ford, and Hyundai Motors (Hyundai), have developed their global expansion rules to secure economies of scale [1] and are running their own specialized strategies. VW chose the strategy building large factories to secure economies of scale, and more than 70% of its overseas factories were built over 300,000 units. Toyota used the strategy of building small factories, and about 47% of Toyota's overseas factories are small-sized with less than 100,000 units. In contrast with VW and Toyota, Hyundai applied a new strategy, which is to build standard-size factories in the overseas market. The factory sizes are standardized into two types, 150,000 units and 300,000 units [2]. In general, it takes about 3–4 years to build an automobile factory, but Hyundai shortened the construction lead time by 50% from 36 months to 18 months to respond to market demands [3,4].

Previous studies mainly focused on the importance of timing and size, but there were lack of discussions on how to reduce the lead times of capacity expansion and analysis of the effectiveness of the strategy in the market, to the best of our knowledge. Therefore, in contrast with the previous studies, this study selected Hyundai as the case to study and used factory size as a fixed variable to determine the optimal timing.

However, the large-scale use of foreign direct investment in manufacturing industry has led to increasingly serious ecological and environmental problems and threatened the sustainable development of manufacturing industry [5]. The factory construction method affects not only the market reaction speed but also the customer's preference for the company [6]. To this end, automobile manufacturers are engaged in visible sustainability activities, including environmentally friendly car production, production of eco-friendly products such as electric vehicles without internal combustion engines, and car recycling, and sustainability reports throughout are published throughout the year. Visible sustainable actions through product development are customer-oriented and depend on the product lifecycle or production cycle. On the other hand, in terms of the business nature of automobiles, invisible sustainable actions need to be prepared on a long-term basis, such as changes in a company's production process, business structure, and global expansion.

ESG (Environment, Social, and Governance) is the key factors to achieve sustainability in business management. In the past, the value of a company was mainly evaluated based on short-term quantitative indicators such as financial statements. However, facing the global climate change crisis and the COVID-19 pandemic, ESG is increasing in importance. Under the paradigm shift, ESG has become a core value directly linked to the long-term survival and prosperity of a company [7]. ESG should be considered as additional evaluation criteria for overseas expansion strategies.

This paper aims to model Hyundai's unique overseas expansion mode as a strategy and analyze the excellence and sustainability. The main contribution of this paper is two-fold. First, we modeled a new capacity expansion strategy, called SPEED which speeds up the lead time ahead of its competitors with. Second, we proposed the procedure and method to evaluate the excellence of the strategy from the perspectives of cost, utilization, and ESG management.

The remainder of this paper is organized as the following; In Section 2, we analyze previous studies on capacity expansion strategy and sustainability. In Section 3, we propose procedure to model and evaluate the strategy. In Section 4, we apply the strategy to the real market with historical demands to drive numerical results. Finally, analytical findings, the contributions of this research, and future works are discussed in Section 5.

## 2. Literature Review

Studies on the capacity expansion problem have been conducted since late 1950s. The strategic models of capacity expansion can be classified into capacity sizing models and/or capacity timing models, as well as plant location and facility type models [8,9]. Research on sizing has been the mainstream in the field of capacity expansion, and the main focus of sizing was sizing methods in terms of demand satisfaction and risk avoidance.

Ref. [10] addressed that the primary objective of capacity expansion is to satisfy demand with minimal investment cost over an infinite horizon. Ref. [11] suggested that the greater risk of running out of capacity the greater the amount investment to avert this contingency. Ref. [12] argued that bigger size of a plant leads to higher probability of failure. Ref. [13] insisted that smaller entry size was better for survival and longevity in new markets. However, designing smaller plants and using step-wise capacity expansions might lead to a decrease in investment risk and an increase in adaptability [14].

The studies on timing of market entry emphasized the strategic decisions that allow shortage or consider the lead time when determining capacity expansion. The two well-known timing strategies of capacity expansion were capacity leading demand and capacity lagging demand [15]. Ref. [16] referred to the three fundamental capacity expansion strategies, lead, lag, and track policies, as the ways to capture market demand. Ref. [10] stated that the service level was guaranteed by timing. In case long lead time is required to construct a new capacity, ref. [17] stated that the timing of capacity investment must be determined with subtle consideration because the decision on the amount of capacity expansion might be changed during the lead time and competitors might force such changes by their actions [18]. Each competing company could invest in capacity either before or after learning about the size of the market demand.

Since then, research on simultaneous optimization of both timing and sizing has been conducted. Ref. [19] considered speed as the basis of competitive advantage in case demands were sensitive to both price and delivery time. Ref. [20] presented a mathematical model and a solution method for determining the optimal quoted lead-time and capacity level for a profit maximizing firm with time-varying and lead-time sensitive demand. Ref. [21] found that early entrants enjoy higher market shares but suffer from lower survival rates than late entrants.

Ref. [22] provided a fundamental approach to cost estimation by conducting a literature review and an empirical study. However, the specific cost data was not disclosed by automakers due to business confidentiality. This paper applied a data-intensive approach [23] and collected all available data of cost values from different reports and

professional magazines (Hyundai IR report, News Papers, Fourin, Automotive news, and Hyundai's previous reports).

In addition to capacity investment, management was deeply involved in capacity utilization. Ref. [9] stated that capacity utilization is a ratio of the actual level of output to a sustainable maximum level of output or is defined as the demand divided by the existing capacity. The capacity utilization ratio level has become an important indicator for the evaluation of the appropriateness of capacity investment and the establishment of an operational strategy.

Another research direction of capacity utilization aimed at maximizing the utilization rate of facilities installed in factories. Ref. [24] proposed a model to determine capacity and its level of utilization for a single machine case in make-to-order manufacturing plants. According to [25], capacity utilization would play a major role in improving profitability compared with other strategic variables including market share, inventory, vertical integration, and industry growth. This inference was further supported by another research stating that excellent capacity management can boost average annual returns on invested capital by as much as 3–4% [26].

The increase in the intensity of environmental regulation changed the effect of capacity utilization on high-quality development in the manufacturing industry. Ref. [27] proposed the problem of overcapacity from the perspective of the ecological environment. Ref. [28] stated that the capacity utilization rate has a significant role in promoting the high-quality economic development of China's regions. The overcapacity would lead to vicious competition in the market, waste of resources, and environmental pollution. Ref. [29] analyzed the transmission mechanism of the impact of overcapacity in manufacturing industry and came to the conclusion that the change of capacity utilization rate has a positive impact on the upgrading of manufacturing industry. The impact of capacity utilization on high-quality development of manufacturing is considered as a threshold variable [30].

For sustainable growth, Ref. [31] proposed the "triple bottom line" composing the economic, social and environmental aspects of sustainability. For sustainable growth, a company should maintain stable long-term economic and socio-ecological performance [32]. Currently, it is difficult to maintain a business unless automakers implement regulations on social safety, environmental regulations, and global greenhouse gas reduction measures [33]. Specifically, there is a robust regulatory action that would have to be fined as much as the number of car sales if the fuel economy was not implemented for car sales, and in the event of a breach of social security, a broad claim would be made from consumers and consumer groups [34].

In contrast to previous studies, this paper proposed a new strategic model to accelerate automotive factory construction to ensure speedy expansion. While expediting market entry by reducing factory set-up time was studied with an analytical approach, we analyzed the factory-based capacity utilization from a managerial perspective by considering who invested in oversea factories newly. Another distinction in this study is that it examines sustainability through environmental and social contributions in relation to the factory building speed and method, as well as the factory's life cycle stages of construction, operation, and closing.

## 3. Materials and Methods

Ref. [35] summarized Management Strategy Evaluation (MSE), which proposed the procedure from identifying concept to interpreting the performance statistics. We interpreted operational rules, transformed management strategy, and simulated and evaluated performance statistics according to the MSE procedure. This study differs from [35] because we took application area and configuration operational condition into account, and applied mathematical programming to compare different strategies. We also used the cases study method to model an empirical case of Hyundai [36–38]. To uphold the capacity expansion rule as a new strategy, we raise three questions:

Q1: What are the innovative points of the SPEED strategy?
Q2: How to design a strategy analysis method in operational research?
Q3: What are the benefits of the SPEED strategy for companies which intend to enter new markets?

To provide answers, we developed a followed a four-step strategy modeling research procedure to answer questions as shown Figure 1.

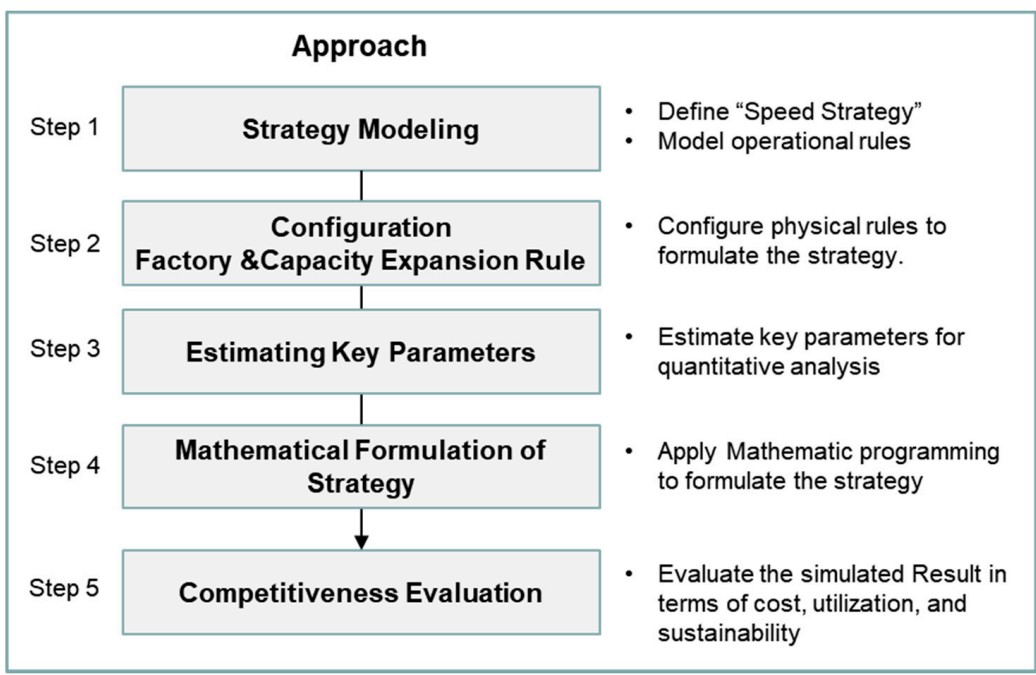

**Figure 1.** Approach to model and evaluate the SPEED strategy.

In step 1, we model the operational rules to interpret the strategic differences by comparing the new and the existing strategies. The characteristics of the new strategy are identified based on decision variable or/and decision way. In step 2, we configure the factory and capacity expansion rules to model the strategy. To acquire common view of the strategy, we configured common physical conditions in factory, and expansion rules. In step 3, we estimate the key parameters for quantitative analysis, which requires consideration of uncertainties of process, parameters, and observation error. We refer to the actual cost structure involving capacity expansion, called cost tree, to estimate the parameters. A data-intensive approach can be used to estimate the key parameters presenting real business conditions. In step 4, the modeled strategy is formulated into a mathematical problem. The process of conducting the simulations and summarizing the results are very time-consuming. The difficult issues at this stage are primarily related to programming development. The strategy is formulated by mathematic programming in this study. Mathematical modeling is the most reasonable approach for comparing the strategies under the same conditions. The competitiveness of the new strategy would be evaluated in cost, demand pattern, utilization, sustainability, and ESG using simulated results in step 5. Ultimately, the selection of a strategy is not only a scientific work but it lies primarily within the view of decision-makers and policy.

## 4. Results

### 4.1. Strategy Modeling

The fundamental differentiator of SPEED strategy was the selection of decision variables. Rather than considering the appropriate factory size that satisfies market demand as a decision variable, the new strategy predefines the standard size and considers construction period as a strategic value to be improved. Since shorter setup time allows more

time to analyze market reactions, time is provided to minimize the risks caused by hasty decision making. It also allows to reflect the most recent risk or environmental change when deciding to build a factory. The strategic concept is shown in Figure 2.

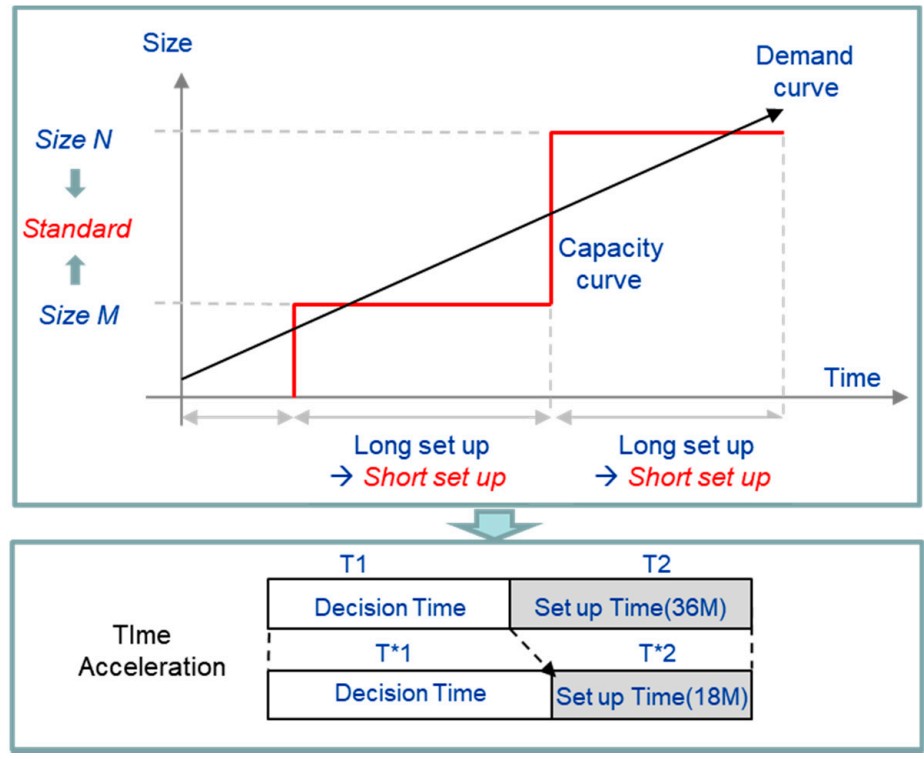

**Figure 2.** Size and timing concept of SPEED strategy.

*4.2. Configuration Factory and Capacity Expansion Rule*

The factory and capacity expansion rules are configured by common sample. As illustrated in Figure 3, the automotive manufacturing factory was comprised by 7 sections: a driving test station, a yard to keep cars temporarily, and five major processes line in manufacturing, including stamping the iron plate, welding the car frame, painting the cars, assembling each part to the car frame, and inspecting the final product.

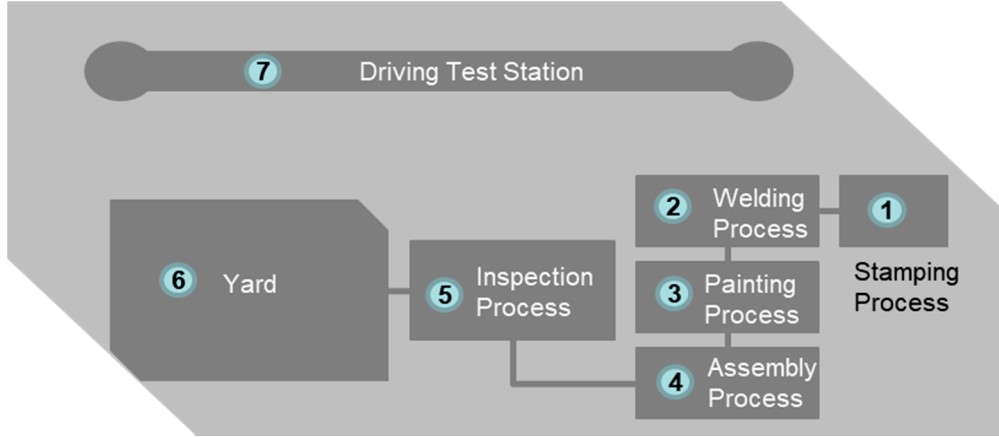

**Figure 3.** Configuration of factory.

The capacity expansion rules can be classified into common processes and individual processes. Common processes take future expansion into account, so larger complex site

space, including a driving test station and a yard, is generally secured. Similarly, stamping, painting, and inspection processes are common processes in a factory that secure capacity with consideration of future line expansion. On the other hand, assembly and welding are individual processes that expanded by line units. As a general concept, additional factories or lines for capacity expansion will be built only if there is no other facility is currently under construction. It is a common practice for automakers to avoid making decisions on building a new factory or line before completing ongoing factory or line constructions. The capacity expansion rules within a factory are presented in Figure 4.

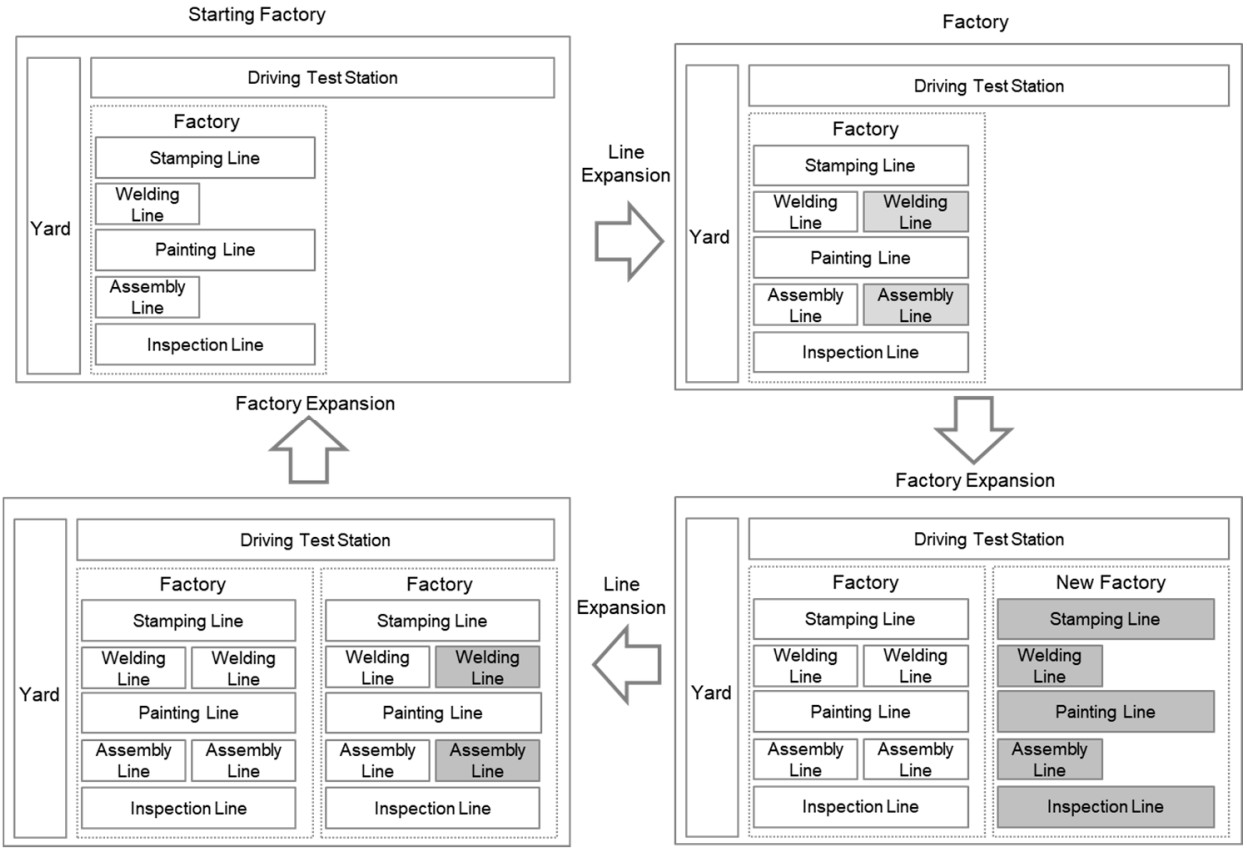

**Figure 4.** Capacity expansion rule.

### 4.3. Estimating Key Parameters

The costs associated with capacity expansion could be classified into three groups: production cost, capacity idles cost, and lost sales cost as shown in Figure 5.

Production costs are derived from the capacity set up cost and Operation and Management (O & M) cost together. The unit production costs also vary depending on the economies of capacity size. The larger the cumulative production volume, the lower the production unit costs. This paper assumed that inventory is not held through pre-production which reduces production costs by expanding production.

Capacity idle cost is the cost due to excessive plant investment or a waste of capacity, and it is calculated as the product of the idle capacity and the unit capacity idle cost. If factory capacity exceeds demand, factory facilities will be temporarily left idle. Once the factory is built, factory operating expenses, including depreciation costs for equipment investment, electricity bills, and maintenance costs, will be incurred annually as fixed costs. In this paper, the unit capacity idle cost is calculated by dividing plant construction cost by actual capacity on the basis of the entire plant investment cost.

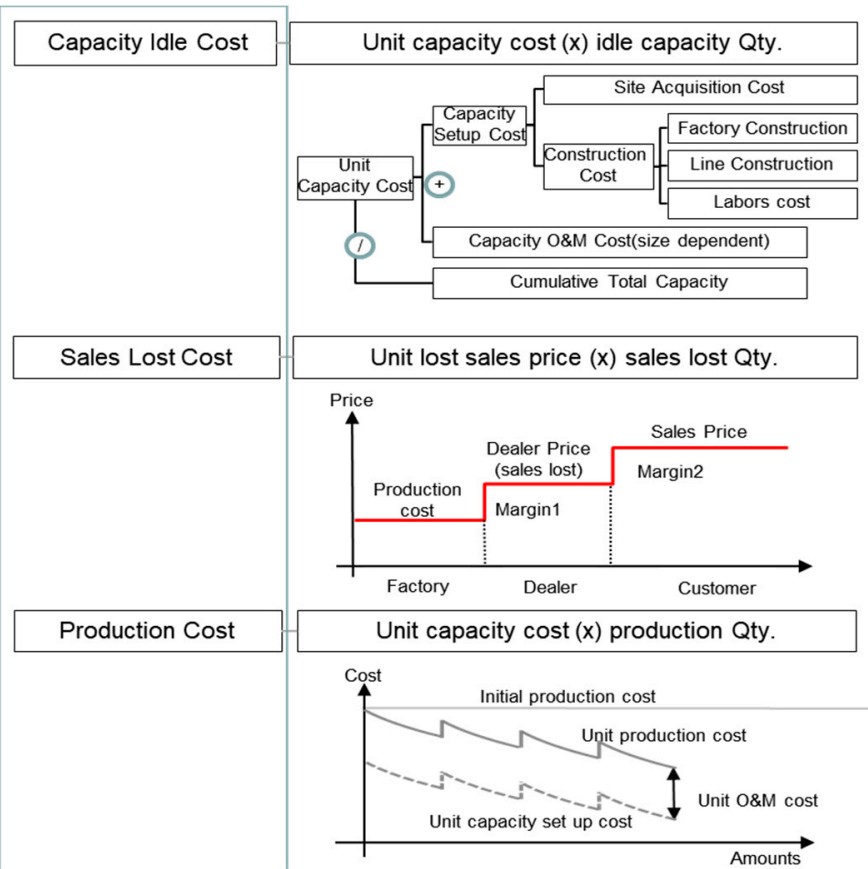

**Figure 5.** Cost structure of capacity expansion model.

Capacity cost is an expense incurred to produce a specific quantity at a specified time, which includes capacity set up cost, and O&M cost. The shorter the construction time, the fewer workers are required to be involved in construction of the factory or line. For example, if the construction of a particular factory lasted for four years, workers were employed for four years. In other words, the speed of factory construction directly related to cost competitiveness.

Lost sales cost refers to the opportunity cost of unmet demand due to lack of capacity, which is calculated as the product of unmet demand and unit product price. The unit product price depends on wholesale price, the price between automakers and dealers. In this paper, the average price of different car models was taken into account since car prices vary by car models. In the calculation of lost sales, we assumed that unmet demand is not back-ordered.

*4.4. Mathematical Formulation of Strategy*

Ref. [9] described several issues in applying operations research methodology to resolve capacity expansion problems. Likewise, this paper faced similar issues and thus we found solutions to formulize each of them.

1st issue: The expansion cost function is usually concave and exhibiting economies-of-scale, while the cost function is piecewise concave. By applying the net increment cost based on time of occurrence and the accumulated capacity, the piecewise concave form of unit capacity expansion cost is realized by different time points in the object function (1).

2nd issue: Expansion size is assumed to be a continuous variable but the number of possible choices of expansion size is limited. To optimize the current technology and operation, the sizes of assembly line-based factories built and operated by operators are limited to 50,000 units, 100,000 units, 150,000 units, 300,000 units, and 600,000 units. In

addition, the production line extension of the factory was carried out according to the rules and restrictions of the existing factory line extension. We applied these conditions to formulation in the constraint (4).

3rd issue: The demand pattern over time is usually partially unknown. So far, we have assumed that the existing capacity exceeds the growing demand at any particular point in time. The strategy was analyzed by utilizing actual market demand to reflect the rapid market growth and the reality of stagnation. In addition, the cost due to the shortage or excess capacity was considered in the objective formula.

4th issue: Most published studies made the assumption that operating costs depend only on the demand volume. However, the operating costs might depend on available technologies and the elapsed time in various applications since the facilities were installed. Automakers operate accounting rules that reflect a certain part of the capacity setup cost as O&M cost. In this paper, we also reflected the O&M cost based on the capacity size applied by actual automakers in the constraint (7).

The indices, parameters and decision variables of the mathematical model are presented as follows.

---

**Definition of Parameters and Decision Variables**

**Indices**
*t: Time period, (t = 1, 2, . . . , T), where T = 15*
*s: Factory and line size option*

**Parameters**
*P: Set of factory size option*
*E: Set of line expansion size option*
*t \*: Time when the existing factory was built*
*tp(s): Construction time of the factory with capacity s*
*te(s): Construction time of line expansion with capacity s*
*Dt: Demand at time t*
*MS: Maximum available capacity per site*
*CP: Factory construction costs per unit capacity*
*CE: Line construction costs per unit capacity*
*$LP_s$: Labor cost for construction of the factory with capacity s*
*$LE_s$: Labor cost for construction of line expansion with capacity s*
*CO: Opportunity cost (sales lost cost) per unit.*
*CA: Cost of new site acquisition*

**Decision Variables**
*$X_t$: Production amount at time t.*
*$Y_t$: Capacity at time t.*
*$ZP_{s1,s2,t*}$: 1 if the factory with s1 size and the line with s2 size is constructed at time t\*. 0: otherwise*
*$ZE_{s3,t,s1,t*}$: 1 if the line with s3 size is constructed at time t within the factory with s1 size constructed at time t\* where $t \geq t^* + tp(s1)$;*
*0, otherwise*
*$CC_t$: Cumulative cost of capacity set up by time t*
*$CM_t$: Cost of factory maintenance and operations at time t*
*$NS_t$ : Number of the acquired site by time t*

---

$$Minimize \sum_{t=1}^{T}(CC_t + CM_t) \cdot \frac{X_t}{Y_t} + \sum_{t=1}^{T}(CC_t + CM_t) \cdot \frac{(Y_t - X_t)}{Y_t} + CO \sum_{t=1}^{T}(D_t - X_t) \quad (1)$$

The objective Equation (1) aims at minimizing production cost, capacity idle cost, and lost sales cost. The first part of the objective function describes production cost, the second part is the description of the idle cost due to excessive capacity investment, and the third part is the cost of lost sales resulting from not meeting demand.

The eight constraint formulas of the capacity expansion problem are defined as follows: Constraint Equation (2) describes constraints on production volume. An automaker

produces cars within the capacity limit to satisfy but not exceed the demand. The automaker does not allow inventories in the factory, and the pulling order system makes production.

$$X_t \leq Min\ \{D_t\ ,\ Y_t\}\ \ \forall t. \tag{2}$$

Constraint Equation (3) describes that the capacity was is extended through the construction of factories and lines. The capacity added or newly established is considered as the possible production capacity at that time. The first part of constraint Equation (3) referred to the cumulative capacity up to the previous time, the second part represents capacity expansion through the new factory construction, and the third part represents capacity expansion through line construction.

$$Y_t = Y_{t-1} + \sum_{s1 \in P} \sum_{s2 \in E} s_2 ZP_{s1,s2,t-tp(s1)} + \sum_{s2 \in E} \sum_{s1 \in P} \sum_{t*=1}^{t-1} s_2 ZE_{s2,t-te(s3),s1,t*}\ \forall t. \tag{3}$$

Constraint Equation (4) describes the rules of production line expansion. Line expansions take place within the capacity limit of the factory, which is referred to the expansion of the automotive production line under the rules of the production line in the factory.

$$\sum_{t=t*}^{T} \sum_{s2 \in E} s_2 ZE_{s2,t,s1,t*} \leq \sum_{s2 \in E} (s_1 - s_2) ZP_{s1,s2,t*}\ \forall s_1 \in P, t^* \tag{4}$$

Constraint Equation (5) prevents concurrent constructions of more than one factory or line. In other words, a new factory or line can only be expanded after the construction of other factory or line is completely. This formula indirectly explained the limitations of resources when expanding factory capacity. If there are unlimited resources, multiple factories can be built and expanded simultaneously. The decision of production capacity expansion is made gradually and carefully while considering resource limitations.

$$\sum_{s2 \in E} \sum_{t'=t-te(s2)+1}^{t} \sum_{s1 \in P} \sum_{t*=1}^{t} ZE_{s2,t',s1,t*} + \sum_{t*=t-tp(s1)+1}^{t} \sum_{s1 \in P} \sum_{s2 \in E} ZP_{s1,s2,t*} \leq 1\ \forall t. \tag{5}$$

Constraint Equation (6) limits the maximum capacity of factories per site. The limit to the number of factories on a site should be followed. When a site exceeded the limit of its capacity to accommodate a factory, an additional site must be purchased at the same time to build a new factory.

$$\sum_{s1 \in P} \sum_{s2 \in E} \sum_{t*=1}^{t} s_1 ZP_{s1,s2,t*} \leq MS \cdot NS\ \forall t. \tag{6}$$

Constraint Equation (7) describes capacity expansion cost. The factory construction costs are composed of land acquisition costs, factory construction costs, labor costs during factory construction, line extension costs, and labor costs during line extension. Longer construction time of a factory will cause an increase in labor costs. The operation and management cost usually accounts for calculated 6% of the total construction costs in the automotive industry, where $CM_t = 6\% \times CC_t$.

$$CC_t = CA \cdot NS_t\ \ + \sum_{t*=1}^{t} \sum_{s1 \in P} \sum_{s2 \in E} (CP \cdot s_1 + CE \cdot s_2 + LP_{s1}) \cdot ZP_{s1,s2,t*}$$
$$+ \sum_{t*=1}^{t} \sum_{t'=t*}^{t} \sum_{s1 \in P} \sum_{s2 \in E} (CE \cdot s_2 + LE_{s2}) \cdot ZE_{s2,t',s1,t*}\ \forall t. \tag{7}$$

The non-negative of all variables, $\tag{8}$

*4.5. Competitiveness Evaluation*

In this study, we compared the SPEED strategy to the most representative strategies of idle capacity avoidance and lost sales avoidance, in order to evaluate the competitiveness of the SPEED strategy. The idle capacity avoidance strategy, benchmarked from Toyota, is an approach to gradually expand small-scale capacity for the purpose of minimizing the risk of excessive investment. The lost sales avoidance strategy, benchmarked from Volkswagen, is an approach that market leaders aggressively enter the market by building large-scale factories from the early stage to dominate the market rather than taking the risk of investment.

4.5.1. Application Data

The historical sales records in 100 markets from 2001–2015 from Automotive News were applied to compare the market competition [39]. The data of market demand in 33 markets, which are divided into introduction (emerging), growing, and matured motorization stages considering CAGR, GDP, and penetration ratio as shown Table 1. Refer to Appendix B for raw data for future research.

**Table 1.** Market and demand analysis.

| Market | Starting Demand ('01) | Current Demand ('15) | CAGR * | GDP ** | Penetration *** | Motorization **** |
|---|---|---|---|---|---|---|
| USA | 2,396,614 | 2,413,534 | 0% | 114 | 797 | |
| UK | 325,617 | 354,085 | 1% | 96 | 519 | |
| Canada | 231,772 | 286,521 | 1% | 105 | 662 | |
| Spain | 187,707 | 130,833 | −2% | 63 | 593 | |
| France | 187,699 | 159,008 | −1% | 93 | 578 | |
| Germany | 175,790 | 170,454 | 0% | 100 | 572 | |
| Australia | 150,832 | 222,491 | 3% | 128 | 740 | |
| Italy | 132,209 | 81,990 | −3% | 74 | 679 | |
| Japan | 112,472 | 96,905 | −1% | 76 | 591 | Matured |
| Netherlands | 73,539 | 55,312 | −2% | 109 | 528 | |
| Belgium | 65,646 | 65,997 | 0% | 100 | 559 | |
| Austria | 53,860 | 56,893 | 0% | 108 | 578 | |
| Switzerland | 44,470 | 45,367 | 0% | 181 | 566 | |
| South Korea | 42,337 | 52,230 | 1% | 59 | 459 | |
| Poland | 41,945 | 47,346 | 1% | 30 | 537 | |
| Sweden | 32,982 | 45,161 | 2% | 123 | 520 | |
| China | 265,469 | 3,640,571 | 19% | 16 | 214 | |
| Russia | 229,824 | 320,000 | 2% | 27 | 293 | |
| Brazil | 228,099 | 379,905 | 3% | 24 | 249 | |
| Mexico | 145,445 | 207,937 | 2% | 23 | 275 | |
| India | 120,322 | 526,297 | 10% | 3 | 167 | |
| Thailand | 105,581 | 270,187 | 6% | 12 | 206 | |
| Iran | 56,340 | 201,511 | 9% | 11 | 213 | Growing |
| Malaysia | 41,157 | 69,050 | 4% | 23 | 361 | |
| Taiwan | 35,445 | 46,514 | 2% | 47 | 322 | |
| Saudi Arabia | 33,952 | 120,933 | 9% | 51 | 336 | |
| Turkey | 29,157 | 149,755 | 12% | 22 | 253 | |
| UAE | 18,287 | 58,952 | 8% | 90 | 313 | |
| Kazakhstan | 2847 | 13,358 | 11% | 26 | 250 | |
| South Africa | 73,770 | 117,805 | 3% | 14 | 165 | |
| Indonesia | 72,314 | 265,433 | 9% | 7 | 68 | Introduction |
| Argentina | 22,040 | 69,568 | 8% | 27 | 165 | (Emerging) |
| Dominican | 2755 | 3200 | 1% | 14 | 163 | |

* CAGR: Compound annual growth rate, ** GDP: Germany is based by 100 (2014), *** Penetration is number of vehicles per 1000 inhabitants (2014 result, Wikipedia), **** Motorization: Introduction(emerging)-growing-matured-decline 4 stage.

The main criteria used for analyzing the strategies were based on the actual operating conditions of automakers. The area of each site to build factories was set to 1 to 4 million square meters in proportion to the capacity size. The site was built in the suburbs and assumed to be $5 per square meter. The land prices were different depending on the level of each market's GDP. The sizes of the unit factories were 50,000 units, 100,000 units, 150,000 units, 300,000 units, and 600,000 units, reflecting the optimal factory size applied by global automakers.

In general, the time required to build a factory was proportional to the size. However, the time could be considered the same as building a minimum unit factory in case of standardization. Standardization made it possible to build a factory in a shorter period. In the case of expanding a factory, according to the rules of expansion, half of the size of the existing factory was expandable, and the time of expansion was set at half the level during the new period.

Labor costs constitute a very large proportion of the actual cost of plant construction. In general, 200 workers and 50 workers are required per month for constructing a factory and a line, respectively [40]. The estimated labor cost is $30,000 per year based on the average labor cost of developing countries, and the amount varies from region to region based on the GDP level of each country.

The capacity setup cost is evenly depreciated over 20 years as a general rule in the manufacturing industry. O & M cost accounts for 6% of the construction cost in practice in the automotive market. The production cost accounts for about 65–70% of the consumer price [41,42]. The data for comparing the SPEED strategy with the idle capacity avoidance and lost sales avoidance strategies were summarized in Table 2.

**Table 2.** Key data for strategy evaluation.

| Classification | | Strategy | | | | | |
|---|---|---|---|---|---|---|---|
| | | Idle Capacity Avoidance | | SPEED | | Lost Sales Avoidance | |
| Capacity size (10k) | Site Size (1 Million Square Meter) | 1 (Small) | | 3 (Mid) | | 4 (Big) | |
| | Site | 20 | | 60 | | 90 | |
| | Factory | 5 | 10 | 15 | 30 | 30 | 60 |
| | Line | 2–5 | 2–10 | 15 | 15–30 | 10–30 | 10–60 |
| Time (year) | Factory construction | 2–3 (short) | | 2 (mid) | | 4 (long) | |
| | Line construction | 1 | | 1 | | 2 | |
| Cost ($1000) | Factory construction | | | 511/unit | | | |
| | Sales lost | | | 14/unit | | | |
| | Site acquisition | | | 5/square meter | | | |
| | Annual labor | | | 3000/labor | | | |

### 4.5.2. Analysis of Competitiveness

(1)  Competitiveness in Cost

The SPEED strategy was cost-competitive in matured and growing markets where more than 100,000 demand sizes were maintained, as shown in Figure 6. The SPEED strategy was superior in 7 out of 16 matured markets (44%) and 4 out of 13 growing markets (30%) and was not superior in emerging markets. The idle capacity avoidance strategy was superior in emerging markets, and the sales lost avoidance strategy was inferior to other strategies in all markets.

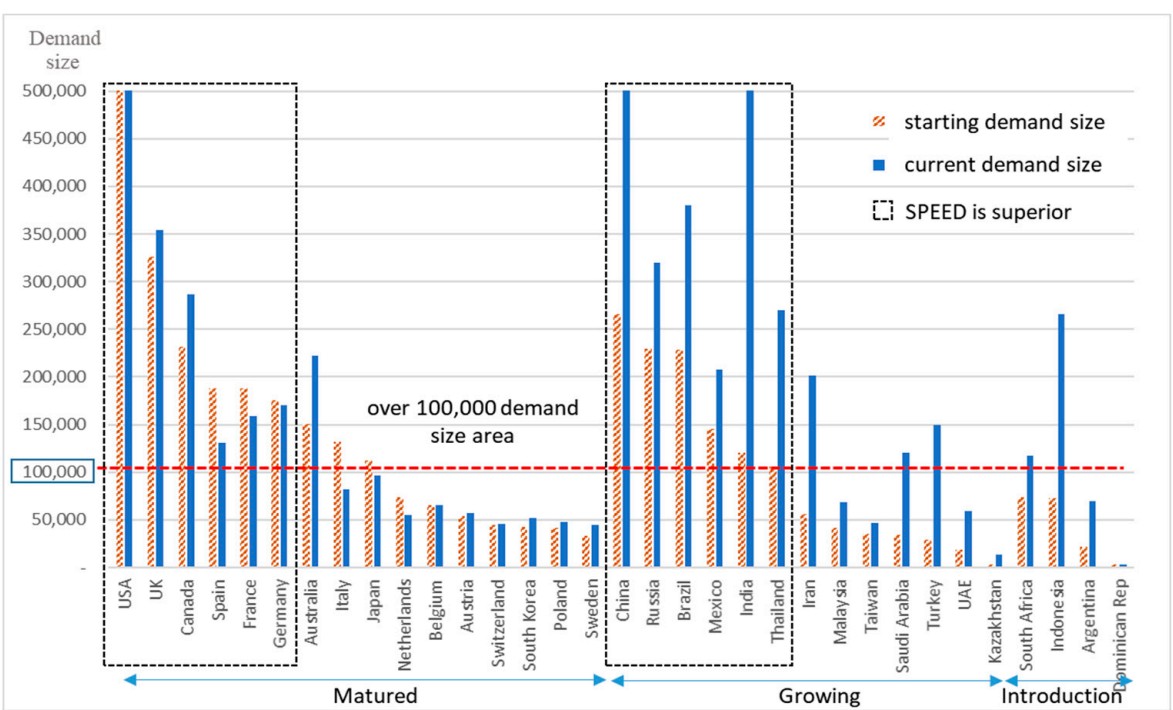

**Figure 6.** Excellence of SPEED strategy for market size.

The superiority of each strategy is different depending on costs and market maturity level. Regarding production cost, the SPEED strategy was superior in 27 out of 33 markets (27%). In regard to market maturity, the SPEED strategy was superior in 14 out of 16 matured markets (87.5%), 9 out of 13 growing markets (69.2%), and 4 out of 4 emerging markets (100%). The idle capacity avoidance strategy was superior in 27 out of 33 markets (88%). By market maturity, it was superior in 12 out of 16 (75%) matured markets, 11 out of 13 growing (85%), and all 4 emerging markets (100%). The lost sales avoidance strategy was superior in 27 out of 33 markets (88%). By market maturity, it was superior in 15 out of 16 (94%) matured markets, 9 out of 13 growing (70%), and 3 out of 4 emerging markets (75%). When applying strategies, it is necessary to diversify the strategy for each market and cost rather than targeting all markets with only one strategy. Refer to Appendix A for more details of the results.

(2)     Competitiveness by Demand Patterns

Strategies were not able to adequately respond to either rapidly increasing or decreasing demands. However, strategies were appropriate for steadily growing demands in a phased and stable manner. The capacity expansion size analysis enabled intuitive interpretation of the strategies by specifying production, capacity idle, and lost sales figures corresponding to demand patterns, as shown in Figure 7.

In the emerging (introduction) stage of the Indonesian market, the SPEED strategy and sales lost avoidance strategy were initially over-invested because the factory size was larger than the initial market size. The idle capacity avoidance strategy (risk avoidance) maintained the optimal capacity scale with a small starting capacity size. In particular, sales lost avoidance strategy (demand satisfaction) with large capacity expansion faces high risks of capacity idle in a stagnant and declining market. In the case of capacity idle, it is necessary to attempt to improve operational efficiency by reducing the actual manpower and production volume. In the growing stage, strategies are differentiated from each other in markets with large demand and rapid growth such as the Chinese market (CAGR 19%) and the Indian market (CAGR 10%).

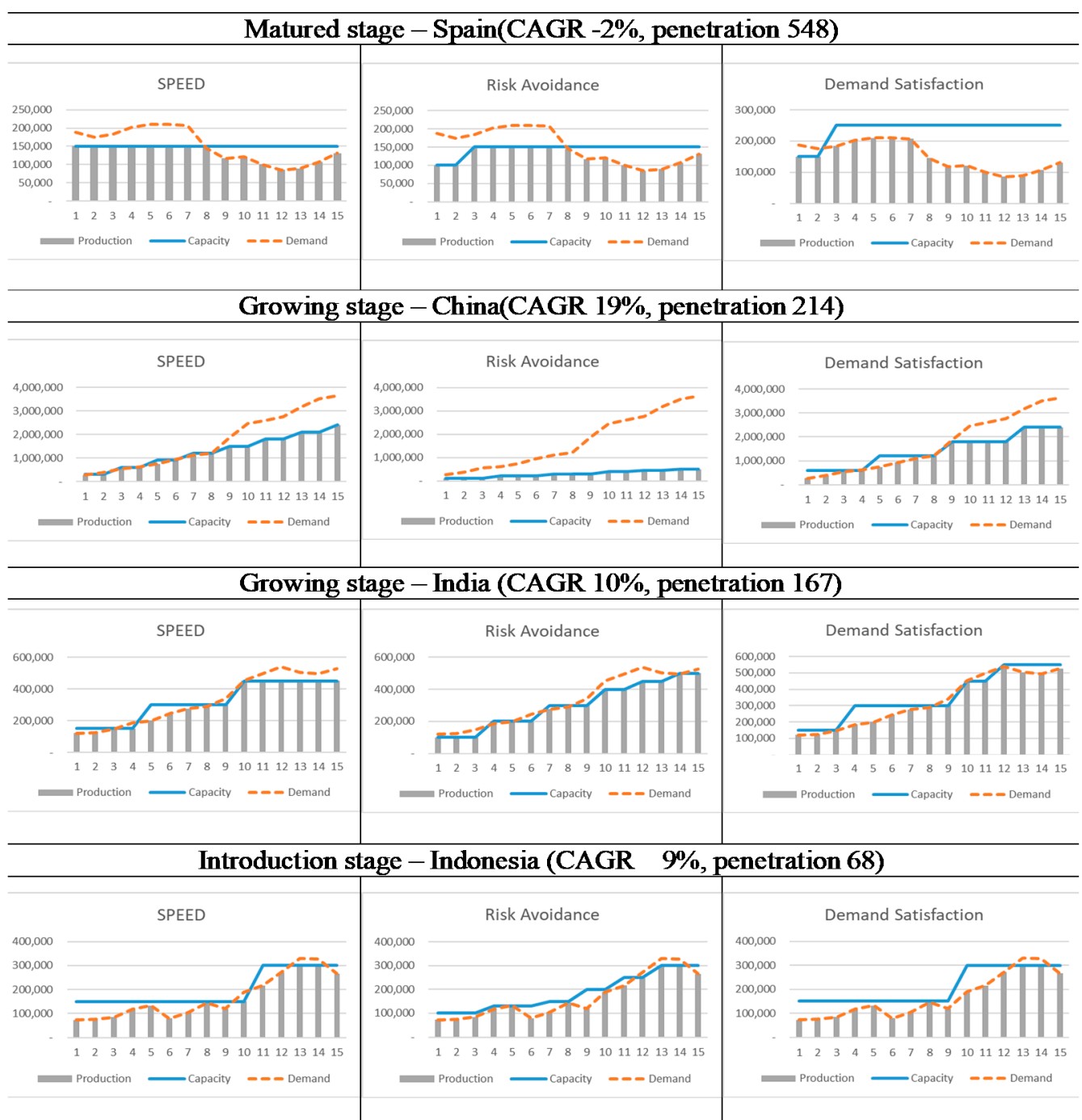

**Figure 7.** Capacity expansion and production volume by demand pattern.

In this paper, considering the budget limitations, we set a rule to prohibit two or more factories from being built at the same time. In the Chinese market, the idle capacity avoidance strategy fails to keep up with the market growth rate and thus sales loss has been rapidly expanding over time. The SPEED strategy and sales lost avoidance strategy show that capacity expansion is in compliance with the growth pace of market demand, while the SPEED strategy also fails to keep up with rapid demand growth as much as demand satisfaction in the Chinese market. In contrast, all strategies responded appropriately to demand growth in the Indian market, where the increase rate is rapid, but the size of demand is not as large as the China market.

(3)    Competitiveness of Utilization

Utilization is closely related to the idle capacity avoidance strategy of minimizing capacity idle cost. However, the SPEED strategy compensates for the weakness of utilization by adjusting the cost of plant construction. In other words, factory standardization plays an important role in preparing for poor utilization by reducing factory construction and O & M costs. The analysis of capacity utilization is important for management after huge investment in foreign markets. Poor utilization of factories means poor operation of investment assets, and thus caused an increase in the cost burden on the company.

Due to the nature of the capacity idle avoidance strategy, which is to gradually expand from the initial small capacity, the utilization ratio remains high even when sales loss occurs. Executives would maximize utilization in respect of return of investment, but lost sales cost and production cost are also considered at the same time in terms of maximizing sales through demand response. The fixed O & M cost is a significant burden on automakers during the lower utilization period. Although the utilization ratio of the SPEED strategy is relatively low comparing to the other strategies, it obtains cost competitiveness in matured and growing markets due to lower capacity idle cost resulted in lower O & M cost.

Ref. [30] showed that capacity utilization, profitability, foreign direct investment, and government participation all have a significant positive impact on the high-quality development in the manufacturing industry. The capacity idle avoidance strategy is closely related to environmental regulations.

(4)    Competitiveness of Sustainability and ESG

ESG factors should be considered throughout the lifecycle of a factory. Environmental factors are considered through construction, operation, and disposal phases. Social factors will be mainly considered in the operation phase. Shorter construction time of the SPEED strategy can reduce air pollution and noise caused by heavy equipment by 50% comparing other strategies. The standardization of the factory can reduce consumption of factory building materials in maximum. In addition, factory standardization leads to process standardization which can broaden the hiring opportunities for local workers with easy training. When retreating business, the materials and equipment of the waste factory can be recycled in other factories. These Environmental (E) and Social (S) effects contribute to the sustainability development of the entering region. The company has the advantages of ESG financial support and ease the local regulation when building overseas factories.

## 5. Discussion and Conclusions

In summary, we justified the value of SPEED as a new strategy. The strategy is motivated by Hyundai's global capacity expansion strategy, which aimed to pursue faster factory construction and size standardization. The SPEED strategy is cost-competitive for entering new markets that have reached a certain scale, such as a matured or a growing market. The standardization of capacity expansion size helps reducing fixed costs even under the circumstance of lower utilization due to market demand fluctuations. In addition, the SPEED strategy makes it possible to overcome the weaknesses of idle capacity avoidance strategy and sales lost avoidance strategy due to its rapid responses to growing demands. From the theoretical point of view, we developed a strategy modeling research procedure and formula to simulate the performance of a strategy.

For an in-depth discussion on the new capacity expansion strategy, the first focus is "What is the innovative point for SPEED strategy". The factory size has been assumed as a decision variable while construction time was fixed. On the contrary, Hyundai considered the setup time as a strategic variable rather than a fixed variable and converted the factory size to a fixed value rather than a decision variable through standardization. This innovative attempt reduced the construction time and demand uncertainty during the construction period of the factory, leading to a reduction on the factory's setup cost. Such a strategic approach is a new challenge that breaks existing assumptions.

The second point of discussion is "How to design a strategy analysis method for operational research?". The optimization technique is a well-known approach to evaluate strategies. We structured and estimated the cost parameters based on real data. The standard factory configuration and line-level capacity expansion rule were modeled to reflect the reality in the formula. Such approaches presented in this paper were proposed as an applicable method to reflect the reality with an operational approach.

The third point of discussion is "What is the effect of the SPEED strategy when firms enter new markets?" A latecomer of a new market should secure differentiated competitiveness through a different approach from the leading companies. In this regard, the SPEED strategy is a highly valuable strategy for latecomers. As a practical example, Hyundai entered the Brazilian market as a latecomer in 2002, 40 years later than Toyota and 49 years later than VW. The market share of Hyundai was 3% with a capacity of 150,000 units, Toyota was 3.1% with a capacity of 140,000 units, and VW was 21% with a capacity of 920,000 units in 2012.

By answering the previous questions, we explained the innovative points and the effectiveness of the SPEED strategy. However, since the study of the capacity expansion strategy in this paper was limited to direct production, it is not applicable to other approaches such as purchase or cooperation. In addition, this study did not consider inter-country product movement arising from factory constructions to respond to demand by the cluster. In the event of building a cell-type factory, new methods are inevitable since the SPEED strategy cannot explain the changing production environment. The key contribution of this paper is that it suggests a new perspective that are different from existing capacity expansion studies of timing and size by modeling the operating strategy in reality.

However new research is still needed to improve the limitations of the currently proposed model and expects for future research. The Hyundai strategy can be modified and refined to capture the more realistic operation applied to many other automotive manufacturing companies by investigating the case practices. As our study focus on the steady implementation of strategy while considering the time. If the strategy can be changed over periods with the advancement of line construction technology, then it would be interesting to study when and which strategy can be selected for cost efficiency. Moreover, we considered the problem of building factories by the market, and we may consider setting up factories by region considering the demands in the adjacent markets. It can address research on capacity expansion problems to optimize global capacity utility. In addition, future research should be conducted to monitor and analyze the specific effect of the SPEED approach in terms of sustainability through actual plant construction and operation.

**Author Contributions:** Conceptualization, G.S.K. and Y.H.L.; methodology, G.S.K.; validation, Y.H.L.; formal analysis, G.S.K.; writing—original draft preparation, G.S.K.; writing—review and editing, G.S.K.; supervision, Y.H.L. All authors have read and agreed to the published version of the manuscript.

**Funding:** This research received no external funding.

**Institutional Review Board Statement:** Not applicable.

**Informed Consent Statement:** Not applicable.

**Conflicts of Interest:** The authors declare no conflict of interest.

# Appendix A

**Table A1.** Result of cost evaluation using mathematical program.

| | Market | Total Cost (1 Mil.) | | | Production Cost (1 Mil.) | | | Capacity Idle Cost (1 Mil.) | | | Lost Sales Cost (1 Mil.) | | | Utilization Ratio | | |
|---|---|---|---|---|---|---|---|---|---|---|---|---|---|---|---|---|
| | | **SPEED** | **Idle *** | **Lost **** | **SPEED** | **Idle** | **Lost** | **SPEED** | **Idle** | **Lost** | **SPEED** | **Idle** | **Lost** | **SPEED** | **Idle** | **Lost** |
| Matured | USA | 17,877.05 | 25,139.29 | 17,850.27 | 7064.79 | 1848.81 | 7727.96 | 17.89 | 0.01 | 182.72 | 10,794.37 | 23,290.48 | 9939.59 | 1.00 | 1.00 | 0.98 |
| | UK | 1711.1 | 2528.94 | 1720.23 | 1389.13 | 1467.18 | 1397.98 | 44.31 | 13.03 | 44.59 | 277.66 | 1048.73 | 277.66 | 0.97 | 0.99 | 0.97 |
| | Canada | 1454.67 | 1874.72 | 1464.72 | 1199.24 | 1392.96 | 1207.52 | 255.44 | 32.02 | 257.2 | 0 | 0 | 0 | 0.82 | 0.98 | 0.82 |
| | Spain | 1023.42 | 1116.08 | 1327.74 | 649.47 | 657.31 | 819.23 | 101.64 | 102.87 | 456.18 | 272.32 | 355.9 | 52.33 | 0.86 | 0.86 | 0.64 |
| | France | 1089.75 | 1162.95 | 1388.94 | 816.38 | 926.34 | 1088.38 | 0 | 64.35 | 162.01 | 273.37 | 172.26 | 138.55 | 1.00 | 0.94 | 0.87 |
| | Germany | 1093.78 | 1174.66 | 1389.69 | 832.36 | 950.49 | 1105.13 | 0 | 48.32 | 162.1 | 261.42 | 175.85 | 122.46 | 1.00 | 0.95 | 0.87 |
| | Australia | 1452.01 | 1318.86 | 1436.37 | 894.08 | 1082.34 | 1180.86 | 0 | 16.98 | 245.22 | 557.93 | 219.54 | 10.3 | 1.00 | 0.98 | 0.83 |
| | Italy | 774.61 | 774.95 | 1110.8 | 564.04 | 535.81 | 808.84 | 210.57 | 38.81 | 301.96 | 0 | 200.33 | 0 | 0.73 | 0.93 | 0.73 |
| | Japan | 778.73 | 645.76 | 1088.22 | 533.46 | 563.87 | 994.9 | 245.27 | 14.96 | 26.39 | 0 | 66.93 | 66.93 | 0.69 | 0.97 | 0.97 |
| | Netherlands | 853.03 | 623.27 | 1098.78 | 368.34 | 550.27 | 711.67 | 484.69 | 49.47 | 387.11 | 0 | 23.53 | 0 | 0.43 | 0.92 | 0.65 |
| | Belgium | 831.94 | 600.51 | 1076.78 | 371.08 | 548.77 | 720.43 | 460.86 | 29.31 | 356.35 | 0 | 22.43 | 0 | 0.45 | 0.95 | 0.67 |
| | Austria | 849.36 | 539.9 | 1094.95 | 327.19 | 442.32 | 632.69 | 522.17 | 0 | 462.25 | 0 | 97.58 | 0 | 0.39 | 1.00 | 0.58 |
| | Switzerland | 1012.02 | 604.98 | 1264.61 | 280.16 | 502.44 | 525.13 | 731.86 | 102.54 | 739.48 | 0 | 0 | 0 | 0.28 | 0.83 | 0.42 |
| | South Korea | 740.42 | 335.25 | 981.33 | 206.3 | 277.68 | 410.14 | 534.12 | 55.7 | 571.19 | 0 | 1.86 | 0 | 0.28 | 0.83 | 0.42 |
| | Poland | 677.47 | 270.43 | 915.67 | 186.23 | 223.01 | 377.56 | 491.24 | 47.42 | 538.11 | 0 | 0 | 0 | 0.27 | 0.82 | 0.41 |
| | Sweden | 882.34 | 475.3 | 1129.35 | 220.47 | 356.3 | 423.29 | 661.87 | 119.01 | 706.06 | 0 | 0 | 0 | 0.25 | 0.75 | 0.37 |

**Table A1.** *Cont.*

| Market | | Total Cost (1 Mil.) | | | Production Cost (1 Mil.) | | | Capacity Idle Cost (1 Mil.) | | | Lost Sales Cost (1 Mil.) | | | Utilization Ratio | | |
|---|---|---|---|---|---|---|---|---|---|---|---|---|---|---|---|---|
| | | SPEED | Idle * | Lost ** | SPEED | Idle | Lost | SPEED | Idle | Lost | SPEED | Idle | Lost | SPEED | Idle | Lost |
| Growing | China | 12,257.8 | 19,416.39 | 12,611.18 | 6322.33 | 1602.35 | 7241.53 | 129.55 | 0 | 666.3 | 5805.92 | 17,814.04 | 4703.35 | 0.97 | 1.00 | 0.89 |
| | Russia | 2240.55 | 3049.7 | 2377.86 | 1823.4 | 1555.5 | 1960.5 | 398.38 | 48.12 | 350.46 | 18.76 | 1446.07 | 66.89 | 0.82 | 0.98 | 0.85 |
| | Brazil | 2064.91 | 2746.5 | 2205.96 | 1745.24 | 1455.68 | 1918.95 | 305.91 | 5.85 | 264.51 | 13.75 | 1284.97 | 22.49 | 0.84 | 1.00 | 0.87 |
| | Mexico | 874.55 | 933.88 | 1175.71 | 643.2 | 754.2 | 848.11 | 17.43 | 60.43 | 319.42 | 213.92 | 119.25 | 8.18 | 0.97 | 0.93 | 0.73 |
| | India | 1795.2 | 1948.49 | 1985.51 | 1403.39 | 1516.78 | 1742.93 | 69.87 | 20.4 | 169.26 | 321.94 | 411.3 | 73.32 | 0.93 | 0.99 | 0.89 |
| | Thailand | 1525.13 | 1614.71 | 1558 | 1211.88 | 1105.74 | 992.26 | 262.69 | 66.83 | 257.41 | 50.56 | 442.14 | 308.33 | 0.81 | 0.95 | 0.79 |
| | Iran | 1122.37 | 961.74 | 1162.48 | 565.83 | 725.72 | 903.65 | 69.59 | 78.7 | 256.25 | 486.96 | 157.32 | 2.58 | 0.89 | 0.89 | 0.78 |
| | Malaysia | 661.86 | 349.22 | 899.39 | 247.66 | 247.68 | 504.82 | 414.2 | 7.14 | 394.57 | 0 | 94.4 | 0 | 0.37 | 0.97 | 0.56 |
| | Taiwan | 715.49 | 310.15 | 955.32 | 184.61 | 237.92 | 369.74 | 530.88 | 70.53 | 585.58 | 0 | 1.7 | 0 | 0.26 | 0.77 | 0.39 |
| | Saudi A. | 723.16 | 544.1 | 1019.84 | 393.33 | 452.39 | 742.5 | 329.83 | 33.46 | 220.82 | 0 | 58.25 | 56.52 | 0.54 | 0.92 | 0.77 |
| | Turkey | 658.76 | 565.87 | 989.96 | 430.61 | 491.15 | 647.1 | 228.15 | 56.58 | 342.86 | 0 | 18.14 | 0 | 0.65 | 0.90 | 0.65 |
| | UAE | 809.94 | 412.71 | 1053.84 | 200.98 | 321.03 | 392.26 | 608.96 | 72.65 | 661.58 | 0 | 19.03 | 0 | 0.25 | 0.81 | 0.37 |
| | Kazakhstan | 668.13 | 204.81 | 905.93 | 31.01 | 71.3 | 63.08 | 637.12 | 133.51 | 842.85 | 0 | 0 | 0 | 0.05 | 0.35 | 0.07 |
| Introduction | South Africa | 640.66 | 587.48 | 971.08 | 447.92 | 461.03 | 678.94 | 192.74 | 43.68 | 292.14 | 0 | 82.78 | 0 | 0.70 | 0.91 | 0.70 |
| | Indonesia | 991.74 | 964.2 | 1153.85 | 746.38 | 781.97 | 852.86 | 165.8 | 115.67 | 254.08 | 79.56 | 66.57 | 46.91 | 0.78 | 0.85 | 0.76 |
| | Argentina | 669.69 | 399.52 | 913.7 | 265.23 | 338.25 | 534.7 | 404.46 | 53.42 | 372.85 | 0 | 7.85 | 6.15 | 0.40 | 0.85 | 0.59 |
| | Dominican | 640.65 | 177.33 | 877.27 | 12.01 | 24.93 | 24.67 | 628.64 | 152.39 | 852.6 | 0 | 0 | 0 | 0.02 | 0.14 | 0.03 |

* Idle: Idle Capacity Avoidance Strategy, ** Lost: Lost Sales Avoidance Strategy.

# Appendix B

**Table A2.** Raw data of global demands.

| Country | Y01 | Y02 | Y03 | Y04 | Y05 | Y06 | Y07 | Y08 | Y09 | Y10 | Y11 | Y12 | Y13 | Y14 | Y15 | Y16 | Y17 | Y18 | Y19 | Y20 | Y21 | Y22 |
|---|---|---|---|---|---|---|---|---|---|---|---|---|---|---|---|---|---|---|---|---|---|---|
| China | 1,659,179 | 2,390,577 | 3,414,827 | 3,837,698 | 4,618,627 | 5,853,807 | 6,972,446 | 7,520,584 | 1,162,0643 | 1,535,9512 | 1,623,5026 | 1,726,0270 | 1,987,8219 | 2,195,3532 | 2,275,3571 | 2,423,4810 | 2,460,3343 | 2,553,0876 | 2,686,7578 | 2,805,1661 | 2,905,1408 | 2,989,9185 |
| United States | 1,711,8669 | 1,681,3068 | 1,663,5535 | 1,686,9894 | 1,695,6222 | 1,656,8453 | 1,615,6717 | 1,324,5391 | 1,043,6926 | 1,159,0357 | 1,277,9036 | 1,449,9305 | 1,560,4014 | 1,651,9157 | 1,723,9526 | 1,751,2392 | 1,782,9663 | 1,754,0546 | 1,730,3303 | 1,713,4510 | 1,711,6044 | 1,701,8554 |
| Japan | 5,623,612 | 5,584,688 | 5,497,036 | 5,541,583 | 5,544,887 | 5,426,394 | 5,099,157 | 4,861,858 | 4,482,483 | 4,812,731 | 4,059,858 | 5,182,158 | 5,155,338 | 5,349,092 | 4,845,245 | 4,885,560 | 4,744,248 | 4,612,275 | 4,551,147 | 4,501,341 | 4,433,499 | 4,364,716 |
| Germany | 3,515,798 | 3,414,441 | 3,404,535 | 3,444,772 | 3,510,472 | 3,658,402 | 3,366,487 | 3,308,597 | 3,969,846 | 3,113,477 | 3,407,008 | 3,301,980 | 3,165,181 | 3,265,126 | 3,409,082 | 3,476,083 | 3,465,091 | 3,432,861 | 3,427,297 | 3,422,760 | 3,430,831 | 3,407,416 |
| India | 707,779 | 730,931 | 868,602 | 1,085,243 | 1,173,000 | 1,436,687 | 1,625,631 | 1,704,030 | 1,992,755 | 2,659,775 | 2,922,312 | 3,161,528 | 2,965,415 | 2,912,208 | 3,095,863 | 3,390,779 | 3,800,267 | 4,192,111 | 4,553,457 | 5,006,998 | 5,455,468 | 5,916,689 |
| United Kingdom | 2,713,472 | 2,830,178 | 2,881,897 | 2,895,746 | 2,762,156 | 2,666,267 | 2,734,931 | 2,420,565 | 2,180,835 | 2,252,760 | 2,201,764 | 2,284,969 | 2,536,909 | 2,796,004 | 2,950,710 | 2,951,956 | 2,810,671 | 2,815,412 | 2,819,711 | 2,789,825 | 2,748,708 | 2,750,181 |
| Brazil | 1,471,604 | 1,391,725 | 1,324,955 | 1,528,468 | 1,620,049 | 1,836,484 | 2,356,494 | 2,664,309 | 3,009,374 | 3,319,413 | 3,412,010 | 3,621,884 | 3,564,804 | 3,312,800 | 2,450,998 | 2,272,704 | 2,396,005 | 2,554,137 | 2,744,920 | 2,942,877 | 3,151,246 | 3,368,465 |
| France | 2,681,420 | 2,543,660 | 2,385,478 | 2,415,480 | 2,482,190 | 2,435,882 | 2,522,558 | 2,505,476 | 2,673,248 | 2,666,241 | 2,630,739 | 2,280,051 | 2,155,360 | 2,165,775 | 2,271,546 | 2,340,088 | 2,391,855 | 2,419,066 | 2,488,420 | 2,397,800 | 2,447,356 | 2,440,480 |
| Canada | 1,534,912 | 1,638,077 | 1,564,669 | 1,521,017 | 1,562,186 | 1,614,898 | 1,700,818 | 1,698,330 | 1,479,314 | 1,572,296 | 1,590,978 | 1,637,781 | 1,728,365 | 1,827,110 | 1,897,492 | 1,902,010 | 1,837,474 | 1,824,882 | 1,827,283 | 1,824,721 | 1,822,502 | 1,813,183 |
| South Korea | 1,411,240 | 1,584,675 | 1,289,594 | 1,067,023 | 1,130,787 | 1,159,667 | 1,228,906 | 1,183,919 | 1,425,535 | 1,524,224 | 1,540,141 | 1,502,897 | 1,504,055 | 1,603,624 | 1,741,009 | 1,719,012 | 1,753,997 | 1,775,994 | 1,796,001 | 1,801,008 | 1,814,994 | 1,825,006 |
| Italy | 2,644,187 | 2,578,771 | 2,456,434 | 2,488,031 | 2,454,927 | 2,553,162 | 2,727,856 | 2,379,356 | 2,359,359 | 2,151,333 | 1,938,613 | 1,517,558 | 1,402,890 | 1,474,807 | 1,639,808 | 1,767,756 | 1,905,476 | 1,978,333 | 2,049,824 | 2,075,123 | 2,050,115 | 2,040,107 |
| Russia | 1,149,121 | 1,128,019 | 1,224,011 | 1,447,373 | 1,612,251 | 1,944,319 | 2,594,045 | 2,952,231 | 1,467,402 | 1,900,538 | 2,669,111 | 2,928,081 | 2,769,812 | 2,481,971 | 1,600,000 | 1,550,005 | 1,700,001 | 2,200,007 | 2,500,005 | 2,750,002 | 2,949,995 | 3,000,000 |
| Mexico | 909,030 | 970,611 | 965,583 | 1,093,371 | 1,141,795 | 1,132,246 | 1,104,698 | 1,032,638 | 751,339 | 818,151 | 900,515 | 982,802 | 1,059,299 | 1,129,450 | 1,299,607 | 1,341,964 | 1,361,928 | 1,381,242 | 1,399,182 | 1,416,621 | 1,433,341 | 1,449,143 |
| Iran | 352,127 | 520,613 | 644,288 | 839,369 | 1,114,282 | 1,177,356 | 1,196,569 | 1,221,838 | 1,469,449 | 1,693,296 | 1,727,321 | 1,066,013 | 791,078 | 1,090,639 | 1,259,441 | 1,353,610 | 1,486,582 | 1,571,063 | 1,618,852 | 1,645,658 | 1,654,884 | 1,676,512 |
| Spain | 1,706,427 | 1,590,009 | 1,672,334 | 1,841,866 | 1,909,651 | 1,904,784 | 1,882,197 | 1,322,705 | 1,061,630 | 1,100,677 | 913,202 | 776,373 | 808,109 | 969,136 | 1,189,389 | 1,138,223 | 1,186,542 | 1,283,315 | 1,402,647 | 1,467,332 | 1,469,869 | 1,407,601 |
| Australia | 754,161 | 807,431 | 899,606 | 938,012 | 967,990 | 944,554 | 1,032,337 | 976,462 | 906,639 | 1,004,713 | 978,580 | 1,079,012 | 1,103,889 | 1,081,655 | 1,112,455 | 1,140,837 | 1,137,814 | 1,143,761 | 1,165,863 | 1,168,456 | 1,167,427 | 1,172,002 |
| Indonesia | 258,264 | 268,206 | 302,724 | 419,338 | 473,935 | 280,753 | 374,018 | 517,348 | 428,220 | 675,224 | 770,729 | 967,214 | 1,174,565 | 1,168,718 | 947,976 | 978,432 | 1,064,666 | 1,136,299 | 1,191,211 | 1,243,027 | 1,296,182 | 1,347,563 |
| Turkey | 182,229 | 160,434 | 357,540 | 683,932 | 701,686 | 605,788 | 574,859 | 477,799 | 546,475 | 747,420 | 854,536 | 765,702 | 841,902 | 755,917 | 935,967 | 871,482 | 920,938 | 1,006,398 | 1,098,527 | 1,064,261 | 1,106,670 | 1,192,917 |
| Saudi Arabia | 212,202 | 231,299 | 295,714 | 325,454 | 401,771 | 436,559 | 487,480 | 562,541 | 546,170 | 630,999 | 601,803 | 678,257 | 710,823 | 771,697 | 755,832 | 758,049 | 770,544 | 779,435 | 796,228 | 825,412 | 843,275 | 844,042 |
| Thailand | 285,353 | 409,209 | 511,050 | 609,230 | 683,698 | 663,333 | 613,521 | 597,202 | 532,891 | 777,994 | 840,342 | 1,303,827 | 1,258,760 | 837,297 | 730,236 | 771,538 | 891,260 | 1,003,249 | 1,112,058 | 1,207,180 | 1,287,102 | 1,355,632 |
| Malaysia | 388,276 | 426,226 | 403,400 | 476,754 | 532,849 | 472,502 | 470,767 | 531,895 | 521,786 | 589,084 | 585,746 | 609,452 | 635,355 | 647,297 | 651,418 | 672,711 | 697,450 | 721,838 | 748,368 | 772,539 | 793,935 | 807,511 |
| South Africa | 368,850 | 354,578 | 379,887 | 468,756 | 599,158 | 693,929 | 642,947 | 504,142 | 379,143 | 477,478 | 554,996 | 606,781 | 625,763 | 620,092 | 589,027 | 605,685 | 658,193 | 681,660 | 714,238 | 733,795 | 751,358 | 766,409 |
| Argentina | 183,663 | 91,504 | 133,725 | 267,836 | 354,593 | 415,824 | 529,271 | 573,788 | 492,714 | 637,092 | 831,215 | 795,782 | 894,629 | 644,495 | 579,732 | 544,979 | 556,176 | 588,535 | 644,704 | 698,284 | 747,209 | 793,289 |
| Belgium | 547,046 | 515,029 | 508,665 | 541,532 | 539,655 | 583,823 | 589,910 | 600,337 | 527,468 | 599,826 | 635,987 | 543,618 | 541,755 | 538,631 | 549,978 | 549,261 | 547,695 | 552,384 | 549,820 | 546,316 | 538,072 | 533,122 |
| Netherlands | 612,826 | 590,407 | 565,460 | 570,207 | 530,831 | 547,993 | 582,880 | 583,668 | 438,017 | 532,607 | 614,357 | 558,850 | 467,540 | 439,608 | 460,934 | 484,895 | 519,584 | 538,154 | 550,610 | 546,432 | 538,654 | 524,569 |
| Taiwan | 322,223 | 357,052 | 376,597 | 438,822 | 473,086 | 328,203 | 315,785 | 207,791 | 272,222 | 299,407 | 356,070 | 347,104 | 355,992 | 404,452 | 422,857 | 432,876 | 443,875 | 453,682 | 461,468 | 467,884 | 478,891 | 491,419 |
| Poland | 349,541 | 328,363 | 383,363 | 353,569 | 270,530 | 280,573 | 347,121 | 377,731 | 361,502 | 373,715 | 319,670 | 311,881 | 331,982 | 369,997 | 394,548 | 401,595 | 415,305 | 426,023 | 444,900 | 468,172 | 498,210 | 530,514 |
| Sweden | 274,849 | 283,642 | 288,810 | 294,500 | 308,985 | 322,574 | 351,154 | 293,005 | 240,905 | 326,746 | 351,422 | 319,243 | 306,969 | 345,989 | 376,339 | 344,509 | 334,252 | 325,759 | 329,338 | 326,629 | 320,974 | 318,309 |
| United Arab Emirates | 114,292 | 115,959 | 123,013 | 154,782 | 180,082 | 216,569 | 275,578 | 323,285 | 199,459 | 210,135 | 225,848 | 281,508 | 333,377 | 367,208 | 368,451 | 365,096 | 369,613 | 377,367 | 392,395 | 410,301 | 407,104 | 413,093 |
| Switzerland | 342,076 | 319,388 | 293,010 | 292,488 | 285,220 | 292,046 | 310,320 | 313,820 | 288,602 | 320,657 | 349,871 | 361,501 | 339,813 | 333,577 | 348,979 | 335,316 | 325,052 | 325,795 | 324,318 | 323,554 | 323,555 | 322,905 |
| Austria | 316,823 | 302,547 | 325,965 | 340,148 | 336,987 | 338,783 | 330,476 | 326,455 | 344,947 | 356,396 | 388,041 | 368,948 | 351,222 | 336,108 | 334,663 | 337,203 | 338,067 | 337,719 | 337,474 | 340,981 | 343,386 | 340,190 |
| Kazakhstan | 23,729 | 25,335 | 26,098 | 47,567 | 32,239 | 35,206 | 45,830 | 31,890 | 9009 | 22,205 | 47,257 | 97,065 | 163,069 | 152,489 | 111,320 | 124,181 | 146,089 | 174,504 | 205,323 | 228,888 | 243,671 | 248,463 |
| Dominican | 18,369 | 26,168 | 7424 | 10,200 | 27,714 | 22,485 | 29,023 | 19,249 | 8260 | 14,229 | 18,476 | 18,792 | 19,299 | 20,178 | 21,330 | 22,493 | 23,577 | 24,585 | 25,517 | 26,328 | 27,122 | 27,982 |

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
