# Peer review of "Speed and Sustainability When Entering New Market"

_sustainability, doi:10.3390/su132212458_

Round 1
Reviewer 1 Report
I liked the research question as well as the methodology used to reach its conclusions. Overall, I believe this is an interesting topic and it has a sound structure. It does a good work on the development of the SPEED strategic approach, based on Hyundai's global capacity expansion strategy, which is to pursue faster factory construction and size standardization. Though, this looks like an interesting paper, some passages are difficult to follow as a result of the poor phrasing and some passages show serious difficulties in taking something that probably sounds good in the native language of the authors into English.
Author Response
Point 1) Though, this looks like an interesting paper, some passages are difficult to follow as a result of the poor phrasing and some passages show serious difficulties in taking something that probably sounds good in the native language of the authors into English.
Respond 1) We checked the English expressions and corrected the sentences with the help of native speakers throughout the paper. (All)
We improved the paper structure by adjusting the numbering(all)
The abstract is improved for clear and easy understanding of the paper. (Line 7 -17)
Literature review is split into separate chapters (Line - 53-213).
Sustainability introduction and strategy interpretation are added (Line 36-43, Line 408-420).
Existing studies and improvements on the strategy evaluation procedure and method were added (Line216-221)
Reference citing was corrected according to the guide. (Line 476-535)

Reviewer 2 Report
Dear Authors,
Thank you for your submission. It is an interesting paper but need further to highlights its real contributions to the current literature. please relate also with the aims and scope of the journal "Sustainability".
Thank you,
M.
Author Response
Point 1) It needs further to highlights its real contributions to the current literature. Please relate also with the aims and scope of the journal "Sustainability".
Respond 1) We checked the English expressions and corrected the sentences with the help of native speakers throughout the paper. (All)
We improved the paper structure by adjusting the numbering(all)
The abstract is improved for clear and easy understanding of the paper. (Line 7 -17)
Literature review is split into separate chapters (Line - 53-213).
Sustainability introduction and strategy interpretation are added (Line 36-43, Line 408-420).
Existing studies and improvements on the strategy evaluation procedure and method were added (Line216-221)
Reference citing was corrected according to the guide. (Line 476-535)
